# Microbial volatile communication in human organotypic lung models

Layla J. Barkal [1,2], Clare L. Procknow[3], Yasmín R. Álvarez-García[4], Mengyao Niu[5], José A. Jiménez-Torres[1,2], Rebecca A. Brockman-Schneider[6], James E. Gern[6], Loren C. Denlinger[7], Ashleigh B. Theberge[8,9], Nancy P. Keller [3,5], Erwin Berthier[8,10] & David J. Beebe[1,2]

We inhale respiratory pathogens continuously, and the subsequent signaling events between host and microbe are complex, ultimately resulting in clearance of the microbe, stable colonization of the host, or active disease. Traditional in vitro methods are ill-equipped to study these critical events in the context of the lung microenvironment. Here we introduce a microscale organotypic model of the human bronchiole for studying pulmonary infection. By leveraging microscale techniques, the model is designed to approximate the structure of the human bronchiole, containing airway, vascular, and extracellular matrix compartments. To complement direct infection of the organotypic bronchiole, we present a clickable extension that facilitates volatile compound communication between microbial populations and the host model. Using *Aspergillus fumigatus*, a respiratory pathogen, we characterize the inflammatory response of the organotypic bronchiole to infection. Finally, we demonstrate multi-kingdom, volatile-mediated communication between the organotypic bronchiole and cultures of *Aspergillus fumigatus* and *Pseudomonas aeruginosa*.

[1] Department of Biomedical Engineering, University of Wisconsin-Madison, Madison, WI 53706, USA. [2] Carbone Cancer Center, University of Wisconsin-Madison, Madison, WI 53705, USA. [3] Department of Bacteriology, University of Wisconsin-Madison, Madison, WI 53706, USA. [4] Department of Chemistry, University of Wisconsin-Madison, Madison, WI 53706, USA. [5] Department of Medical Microbiology and Immunology, University of Wisconsin-Madison, Madison, WI 53706, USA. [6] Department of Pediatrics, School of Medicine and Public Health, University of Wisconsin-Madison, Madison, WI 53705, USA. [7] Department of Medicine, School of Medicine and Public Health, University of Wisconsin-Madison, Madison, WI 53705, USA. [8] Department of Chemistry, University of Washington, Seattle, WA 98195, USA. [9] Department of Urology, University of Washington School of Medicine, Seattle, WA 98195, USA. [10] Tasso Inc., Seattle, WA 98119, USA. Correspondence and requests for materials should be addressed to E.B. (email: erwin.berthier@gmail.com) or to D.J.B. (email: djbeebe@wisc.edu)

Acute respiratory infections contribute to significant global morbidity and mortality, especially in young children and the immunocompromised[1, 2]. Much of our inability to treat these infections can be attributed to a limited understanding of the critical events that occur when microbes, or microbial compounds, are inhaled. Parsing these events, however, is challenging due to the incredibly complex lung environment; physically, there is a gradient of airway shape from trachea to alveoli as well as an air-liquid interface and cyclic changes in airflow and airway pressure. Biologically, the types of epithelial cells that line the airway change along the respiratory tree, and there are immune cells distributed throughout, ready to respond. Also, we now know that even healthy lungs harbor a resident microbiome containing bacteria, fungi, and viruses[3, 4]. With such a complicated context for lung infection, efforts to study critical pathogenic events of inhaled microbes are complicated by available tools that cannot capture much of the microenvironmental complexity or that cannot support the multifaceted analyses needed to describe these critical events.

Traditional in vitro methods tend to be simple, inexpensive representations of the lung that have also led to promising new insights, but can oversimplify the infection context and therefore may misrepresent or miss entirely some of the critical pathogenic events. Many in vitro models of pulmonary infection represent infection by placing pathogens directly atop the air-exposed apical surface of a monolayer culture of pulmonary epithelial cells, investigating the effects of direct communication between respiratory epithelium and pathogen[5], and migration through the epithelial layer[6]. This approach is appealing and achieved in Transwell systems or more recently in micro-engineered lung-on-a-chip systems, both of which utilize a microporous membrane as a support for the epithelial layer, and thus do not integrate fibroblasts or supporting matrix[7, 8]. Additionally, these systems do not support investigating the effects of communication between host and microbes via volatile compounds, which are known to modulate biology of the pathogens and uniquely identify infections of the host[9–11]. There is a need for new in vitro methods that better model the complexity of the in vivo lung microenvironment, that support different modes of pathogen interaction, and that enable high-content, orthogonal analyses.

Here we present a microscale in vitro platform that models the structure of the human terminal bronchiole, the narrowest conducting airway in the lung, and that can be used to investigate both physical contact and volatile communication between host and pathogen(s) at this level of the respiratory tree. To build the organotypic model, we leverage a microscale approach to reduce cell requirements, which allows us to use primary human cells, and enable a more structurally and geometrically accurate tissue model including stromal components. The organotypic bronchiole also supports integrated assays, including tracking of immune cell recruitment. We use the fungus *Aspergillus fumigatus* as a model inhaled fungal pathogen, and show that direct infection of the bronchiole model causes an inflammatory cytokine response and induces prompt leukocyte extravasation and recruitment. Finally, we describe a modular multikingdom culture method that enables volatile communication between the bronchiole model and cultures of two other microbes. This platform extends our ability to understand multikingdom interactions, and we report that volatile communication between the human bronchiole model and the fungus *A. fumigatus* is modulated by volatile contact with *Pseudomonas aeruginosa*, an often co-inhabiting bacterium in patients with cystic fibrosis.

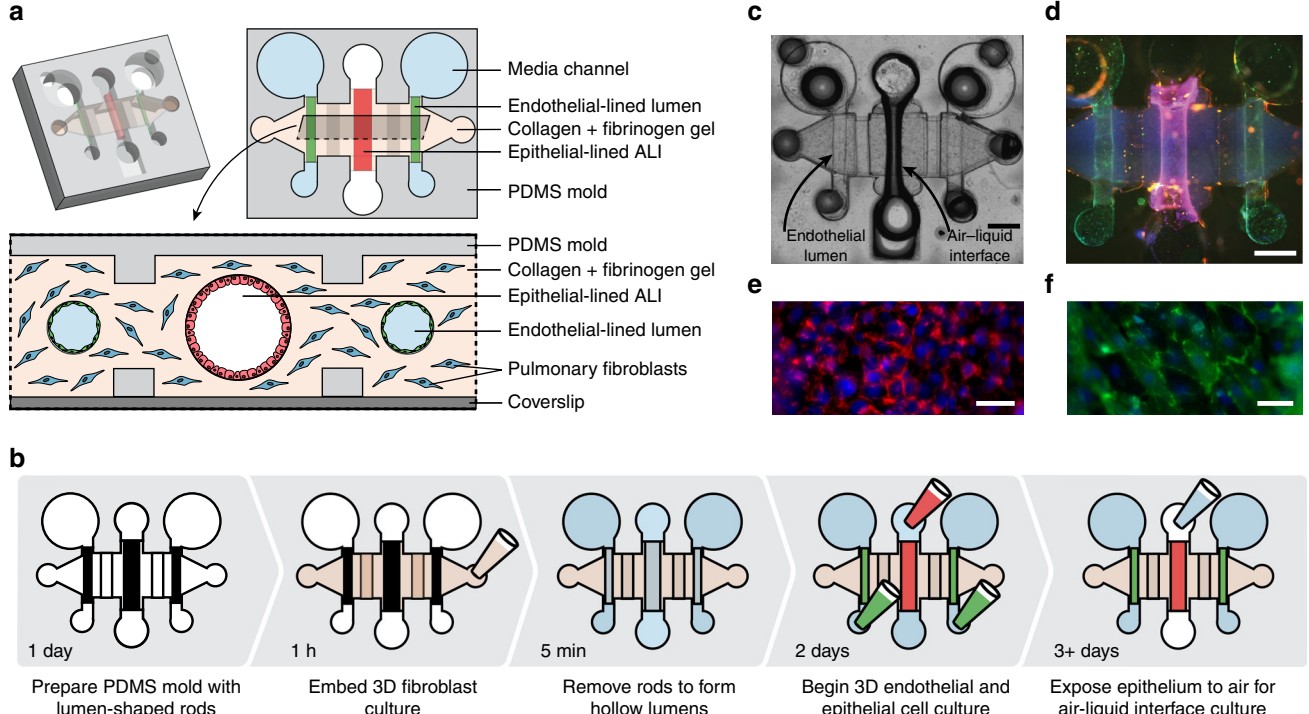

**Fig. 1** Cell culture in the organotypic bronchiole model. **a** Schematic of the organotypic bronchiole device, including an isometric view, a horizontal section, and a vertical section. **b** Steps, and their associated duration, to set up organotypic bronchiole devices. **c** Photo of one organotypic bronchiole device with air in the center, epithelial cell-lined lumen and media in the two side, endothelial cell-lined lumens. Scale bar is 500 μm. **d** Organotypic bronchiole model stained with Hoechst (blue, nuclear stain), anti-CD31 antibody (green, endothelial tight junction marker), and anti-EpCAM antibody (red, epithelial cell–cell adhesion maker). Scale bar is 500 μm. **e** Magnified image of the epithelial cell monolayer present in the center lumen from **d**, scale bar is 200 μm. **f** Magnified image of the endothelial cell monolayer present in the side lumens from **d**. Scale bar is 200 μm

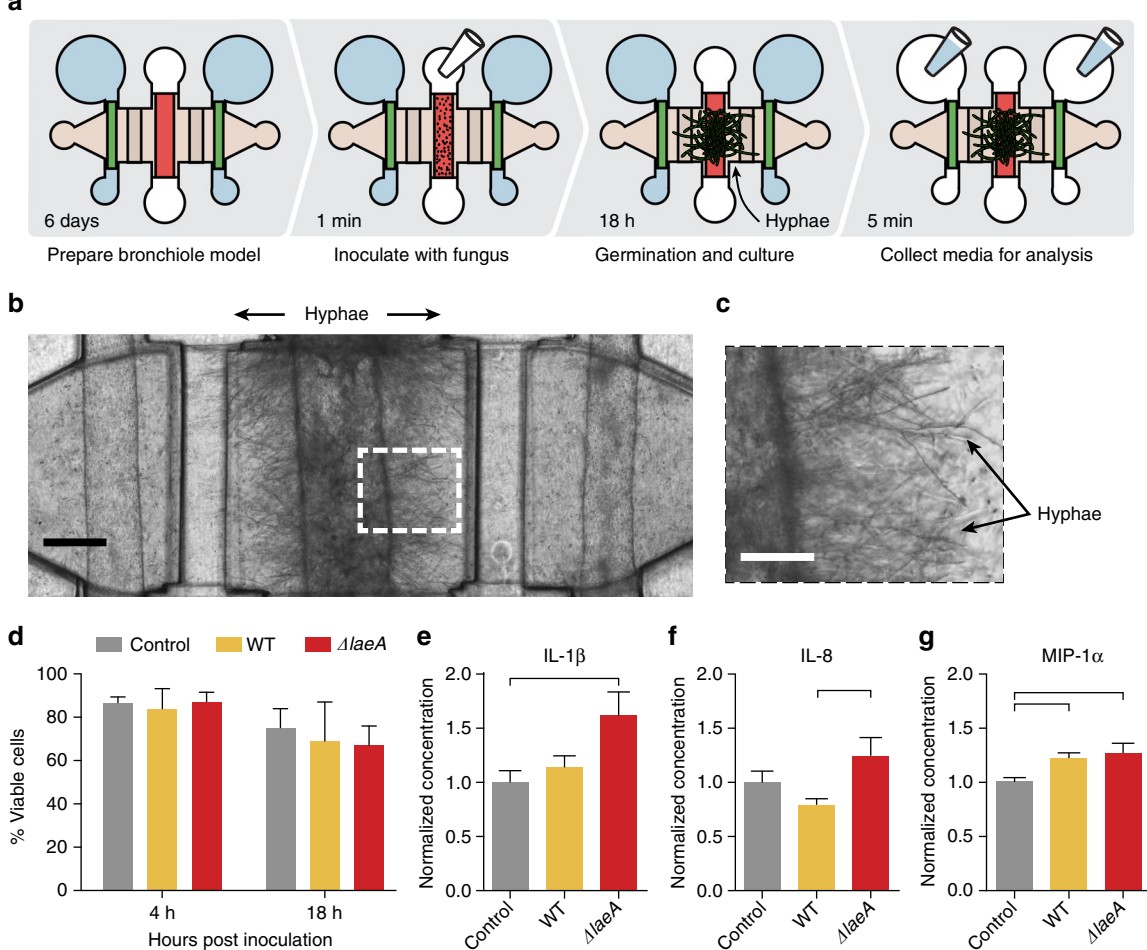

**Fig. 2** Direct fungal contact stimulates cytokine response in the organotypic bronchiole model of invasive pulmonary aspergillosis. **a** Experimental overview depicting how the bronchiole models are inoculated with spores, how the spores germinate and grow into hyphae, and how media is collected from the models for chemical analysis. **b** Photo of bronchiole model 18 h after inoculation of WT *A. fumigatus* spores directly into the center, epithelial cell-lined lumen. Fungal hyphae extend through the epithelial barrier into the surrounding matrix. Scale bar is 250 μm. **c** Closer image of fungal hyphae in **b** protruding from the center lumen. Scale bar is 100 μm. **d** Cell viability analysis of center, epithelial cell-lined lumens at 4 and 18 h post inoculation with spores. Conditions were performed in triplicate with the mean value plotted as a bar and error bars representing standard error of the mean. Data were analyzed using an ordinary two-way ANOVA with Tukey's multiple comparisons test; none of the comparisons met the significance threshold of a *p*-value < 0.05. **e–g** Cytokine concentrations measured in media collected from the side, endothelial cell-lined lumens of bronchiole models inoculated with WT *A. fumigatus* spores or *ΔlaeA A. fumigatus* spores normalized to control, non-inoculated bronchiole models. Each plotted bar represents the mean of nine organotypic devices prepared on three separate days and error bars represent standard error of the mean. Data were analyzed using an ordinary one-way ANOVA with Tukey's multiple comparisons test and horizontal brackets denote comparisons between conditions that are statistically significant (*p*-value < 0.05). Additional cytokines from this experiment are presented in Supplementary Fig. 3

## Results

**Organotypic bronchiole design and operation**. We engineered a microscale organotypic model of the human bronchiole that supports functional readouts of cellular behavior in the context of infection. The model is comprises three cell-lined lumens within a 3D matrix of collagen and pulmonary fibroblasts (Fig. 1a, Supplementary Data 1). The center lumen is lined with primary human bronchial epithelial cells and can be filled with air to form an air–liquid interface (ALI) culture representing the in vivo airway (Fig. 1b, c). The two flanking lumens form the vascular compartment and are lined with primary human lung microvascular cells. Each cellular component of the organotypic bronchiole model plays a role in mimicking in vivo function: the fibroblasts contained within the collagen gel provide support for healthy monolayers of endothelial and epithelial cells, which form physical barriers between the luminal and matrix spaces. In vivo, these cells form monolayer barriers that regulate the passage of

compounds, fluid, and pathogens. Endothelial cell monolayers express endothelial junction protein CD31 (Fig. 1d, f) and form functional barriers capable of containing a labeled 4 kDa dextran (Supplementary Fig. 2a–c) by day 3 of culture in the organotypic bronchiole model. The airway epithelial cell monolayers express cell–cell adhesion molecule EpCAM after 3 days of culture at the ALI (Fig. 1d, e) and also function as barriers that can contain the labeled dextran (Supplementary Fig. 2d, e). With tight barrier formation and the triple lumen geometry of the device, the organotypic bronchiole model mimics the vascular, extracellular matrix, and airway compartments of the in vivo bronchiole.

To facilitate modeling infection and immune responses in the organotypic bronchiole, each lumen has separate pipet-accessible ports (Fig. 1a, b). This allows suspensions of microbes to be added to the airway lumen and peripheral immune cells to be added to the vascular lumens. The ports also allow for simple removal and analysis of samples from each compartment separately. Despite

the small culture size and low volume of conditioned media contained within each lumen, there is sufficient material recovered for chemical analysis with bead-based ELISA. The cellular and extracellular matrix components of the organotypic bronchiole model can also be removed and processed histologically as a tissue sample (Supplementary Fig. 1). In addition to compatibility with off-chip biochemical and histological readouts, the bronchiole model is constructed atop a glass coverslip, which facilitates time-lapse imaging of infection progression and immune cell response.

The microscale nature of the organotypic bronchiole model is particularly enabling for mimicking the in vivo lung. Perhaps most importantly, operating at microscale enables the use of primary human cells; bronchial epithelial cell requirements are ~5× less than typical transwell ALI cultures[12], which allows the organotypic bronchiole device to be used to investigate a broader experimental space than is currently feasible in macroscale. Additionally, microfabrication techniques allow for quick fabrication of device arrays and offer extensive control over the geometry of the bronchiole model, especially the ability to create luminal structures within a hydrogel matrix[13]. Cells grown in lumen-shaped monolayers are biologically distinct from those grown in flat monolayers, and with new fabrication methods, lumens can be created simply and reproducibly[14, 15]. These methods offer significant control over lumen geometry, and the lumens in the organotypic bronchiole model were designed with specific shapes, sizes, and orientations to best match the in vivo structure of the human lung; center bronchiole and side vascular lumens lie in parallel as airways and vessels travel together in the lung, while diameters were chosen to achieve cross-sectional areas of approximately 0.3 and 0.07 mm$^2$, respectively, to match average human measurements[16, 17].

**Organotypic bronchiole cytokine response to fungal infection.** The terminal bronchioles in the human lung employ varied methods of defense from inhaled pathogens. Besides physical removal of inhaled pathogens by mucociliary flow, the respiratory epithelial cells that line the airway play an active role in defense by secreting factors apically, including β-defensins, lactoferrin, lysozyme, and antimicrobial peptides to directly attack microbes[18]. They also secrete factors basally to recruit immune cells to the site of invasion[19, 20]. Stromal cells in the lung including pulmonary fibroblasts and pulmonary microvascular cells can also combat microbial pathogens by secreting immunomodulatory factors[21–23]. Together, this creates a complex 3D inflammatory milieu that is still incompletely understood. Using the ubiquitous human fungal pathogen *Aspergillus fumigatus*, which causes invasive and often fatal disease[1], we show that the organotypic bronchiole model allows us to measure an integrated chemical response to infection. We observe the model's response to infection with both WT *A. fumigatus* and a *ΔlaeA* knockout *A. fumigatus* strain. LaeA regulates production of immunomodulatory secondary metabolites, some of which elicit an immune response and some that help evade the immune response[24–27]. The knockout strain has been shown to be less pathogenic than WT in both murine and zebrafish models of invasive aspergillosis[24, 28].

Fungal spores are introduced in suspension into the center airway lumen of complete bronchiole models and after 18 h of culture, the germinated fungus forms a dense mat of hyphae (Fig. 2a). The hyphae punch through the walls of the airway lumen and extend nearly to the flanking microvascular lumens (Fig. 2b, c) in a manner consistent with invasive aspergillosis in humans. Epithelial cell viability is measured at 4 and 18 h post infection (Fig. 2d) as cell death at the site of infection can

facilitate an immune response. In the organotypic bronchiole model, fungal inoculation does not alter epithelial cell viability over the first 18 h of infection (Fig. 2d). To characterize the integrated cytokine response of the organotypic bronchiole model to this infection, conditioned media is recovered from the vascular lumens and inflammatory cytokines measured (Fig. 2a). The inflammatory cytokine Interleukin-1β (IL-1β) is induced in bronchiole models inoculated with *ΔlaeA* spores compared with non-inoculated controls, but WT spores do not induce production to the same degree (Fig. 2e). Interleukin-8 (IL-8) demonstrates a similar pattern of induction to IL-1β: *ΔlaeA A. fumigatus* induces production of IL-8 but WT *A. fumigatus* does not (Fig. 2f). Macrophage Inflammatory Protein-1α (MIP-1α), on the other hand, is induced in both *ΔlaeA* and WT-inoculated models compared to non-inoculated control bronchiole models (Fig. 2g). Other cytokines measured in these experiments are presented in Supplementary Fig. 3.

By sampling conditioned media from the side vascular lumens, we measure the integrated response of the organotypic bronchiole model to direct, invasive fungal infection; the basolateral epithelial response, the fibroblast response, and the apical endothelial response are aggregated into the net, functional signal that would be presented to circulating immune cells. Of note, bronchiole models inoculated with *ΔlaeA A. fumigatus* spores had stronger induction of proinflammatory cytokines (IL-1β, IL-8) than models inoculated with WT spores. The stronger cytokine induction in *ΔlaeA* conditions could be due to loss of production of LaeA-regulated fungal secondary metabolites that normally act to suppress the host immune system[25, 26, 29]. Alternatively, it could be due to LaeA-regulated changes in spore surface proteins that make the fungus more readily recognized by host cells[27, 30]. In either case, the greater inflammatory cytokine response to *ΔlaeA A. fumigatus* than WT *A. fumigatus* may explain why, in animal models, the *ΔlaeA* strain is more readily cleared and animals have higher survival rates: with a greater immune response, the host can better defend against this knockout strain[24]. Also of note is the expression of MIP-1α, which is increased in both WT and *ΔlaeA* fungal conditions (Fig. 2g). MIP-1α can be produced by both pulmonary fibroblasts and respiratory epithelial cells[31, 32], among other cell types, so this cytokine level is truly an aggregate response of the organotypic bronchiole model to fungal infection. Induction of MIP-1α in both fungal conditions supports the observation in animal models that MIP-1α plays a critical role in defending neutropenic hosts from invasive pulmonary aspergillosis, primarily by recruiting lymphocytes[33].

**Leukocytes extravasate and migrate to fungal infection.** Leukocyte recruitment is a central event in the innate immune response to fungal pathogens. Both microbial-derived compounds and inflammatory cytokines secreted by host tissue at the site of infection are sensed by neutrophils, macrophages and eosinophils, which migrate to the invading pathogen and attack[34–36]. Conversely, there are fungal strategies to combat immune cell recruitment, including production of fungal secondary metabolites that have been shown to suppress polymorphonuclear leukocyte (PMN) migration[26]. Therefore, production of inflammatory cytokines, as described above, offers an incomplete picture of the host response to fungal infection, and there is a need for more functional characterization of the immune cell migration response.

In the organotypic bronchiole model, immune recruitment to direct, invasive infection is assessed by inoculating the center, epithelial lumen with fungal spores. Over the course of 18 h of incubation, the spores germinate and hyphae cross through the

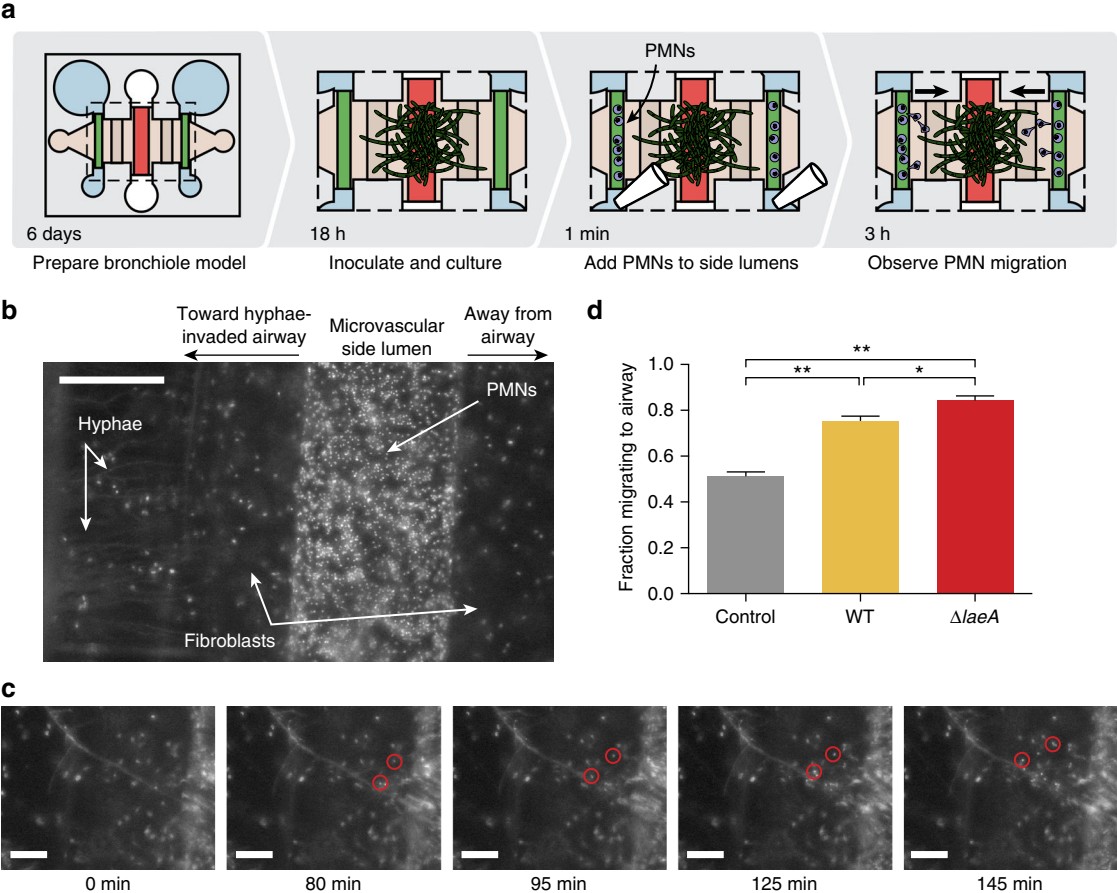

**Fig. 3** Polymorphonuclear leukocytes (PMNs) preferentially migrate toward fungal hyphae. **a** Diagram of the experimental setup. Fungal hyphae extend from the center, epithelial cell-lined lumen 18 h after inoculation with WT or *ΔlaeA A. fumigatus* spores at which point PMNs purified from whole blood are added to side, endothelial-lined lumens and migration is observed. **b** Magnified image of one side lumen filled with PMNs, all cells stained with Hoechst (nuclear stain). Scale bar is 200 µm. **c** Time lapse frames tracking two PMNs (red circles) as they leave the side lumen and migrate toward the hyphae extending from the center airway. Frame time is given in minutes, and the scale bars are 100 µm. **d** The fraction of PMNs that extravasated from the airway-facing side of the endothelial lumens and migrated in the direction of the center, hyphae-invaded airway lumen, compared with those that extravasated and migrated in the opposite direction. Each plotted bar represents the mean of nine organotypic devices prepared on three separate days, with fractions calculated as the average of both endothelial side lumens per device. Error bars represent standard error of the mean. Data were analyzed using an ordinary one-way ANOVA with Tukey's multiple comparisons test: *denotes *p*-value < 0.05 and **denotes *p*-value < 0.005

epithelial cell layer to extend toward the side, microvascular lumens (Fig. 3a). Purified white blood cells, specifically PMNs, are added to the microvascular lumens, where they travel along the endothelial surface, extravasate through the endothelial cell layer, and migrate through the collagen matrix to reach invading hyphae (Fig. 3a, b). Time-lapse imaging of the side lumens enables analysis of PMN extravasation and migration at a single cell level (Fig. 3c, Supplementary Movies 1 and 2); in Fig. 3c, PMNs are contained within the side lumen at the beginning of the experiment (0 min), two PMNS are highlighted in red once they extravasate (80 min), and are tracked as they migrate through the matrix to reach the fungal hyphae (145 min) where they remain for the rest of the experiment. To quantify the PMN response to non-inoculated center lumens, and those inoculated with WT and *ΔlaeA A. fumigatus*, we look at the PMNs that extravasate from the side lumen and measure the proportion that migrate toward the center lumen and hyphae. In control, non-inoculated lumens, the proportion is 50%, indicating no preferential migration of the PMNs toward the center lumen (Fig. 3d). WT *A. fumigatus* induces more directional migration, and *ΔlaeA A. fumigatus* prompts 80% of migrating PMNs to head toward the center

lumen and associated hyphae (Fig. 3d). Models of infection with spores or fungal germlings rather than established hyphae resulted in little directional PMN migration (Supplementary Fig. 4).

Migration of immune cells provides a functional assessment of the inflammatory milieu as a whole. In the organotypic bronchiole model, migration is an integrated response to signals secreted from the endothelial cells, the embedded fibroblasts, and the basolateral surface of the epithelial cell monolayer. Additionally, fungal-derived compounds can also stimulate immune cell recruitment[37]. Together, these signals form chemical gradients sufficient to stimulate PMN extravasation through a layer of microvascular endothelial cells and directed migration through the 3D collagen matrix to invading fungal hyphae. Both WT and *ΔlaeA A. fumigatus*-infected airway lumens induce PMNs that extravasate from the microvascular lumens to migrate directionally toward the infected airway lumen (Fig. 3d). While our goal was to observe the functional PMN response to the full milieu, rather than to pinpoint specific chemokines responsible for migration, it is useful to consider the chemokines measured in the related experiment shown in Fig. 2. Interestingly, WT infection

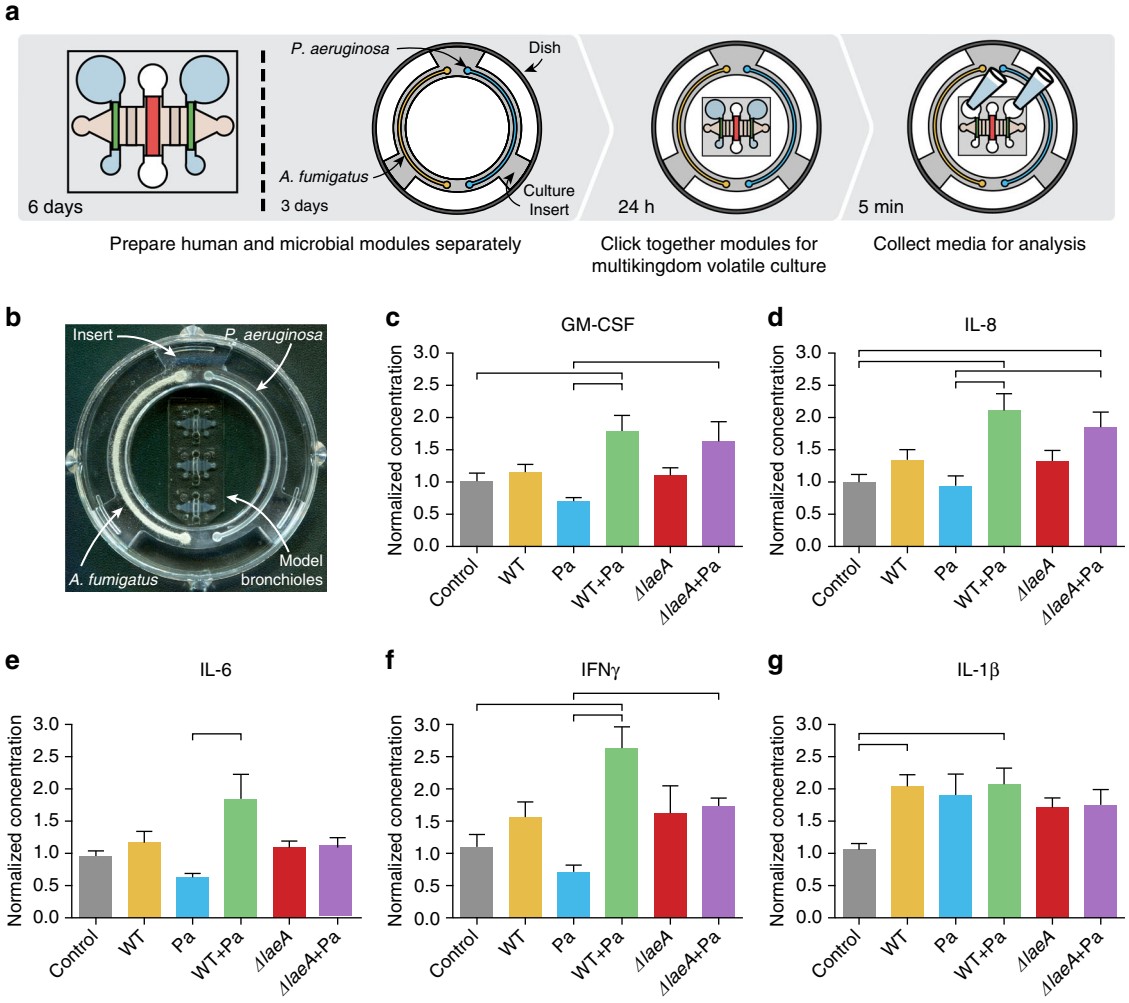

**Fig. 4** Microbial volatiles impact epithelial cell secretion of inflammatory cytokines. **a** Key experimental steps. **b** Photo of the experimental setup. The microbial culture insert is inoculated with *A. fumigatus* on the left and *P. aeruginosa* on the right. The insert sits within a securely covered 50 mm glass bottom dish containing three organotypic bronchiole devices, thus facilitating volatile factor contact between the microbial cultures and air-exposed center lumens lined with bronchiolar epithelial cells. **c–g** Cytokine levels in conditioned media from the side, endothelial-lined lumens measured after 24 h of volatile coculture with an insert all non-inoculated, half WT *A. fumigatus*, half *P. aeruginosa*, half each WT *A. fumigatus* and *P. aeruginosa*, half Δ*laeA A. fumigatus*, and half each Δ*laeA A. fumigatus* and *P. aeruginosa*. Each plotted bar represents the mean of nine organotypic devices prepared on three separate days and error bars represent standard error of the mean. Data were analyzed using an ordinary one-way ANOVA with Tukey's multiple comparisons test and horizontal brackets denote comparisons between conditions that are statistically significant (*p*-value < 0.05). Additional cytokines from this experiment are presented in Supplementary Fig. 6

does not induce production of the canonical chemokines IL-1β and IL-8 at this time during the infection (Fig. 2e, f); MIP-1α (Fig. 2g) or other signals outside of our screening assay may be responsible for the PMN recruitment to the site of infection. Further, it is important to consider that some fungal metabolites can inhibit PMN migration, notably the *A. fumigatus* secondary metabolite endocrocin[26]. LaeA loss results in decreased endocrocin production[38], and may explain, in part, the increase in PMN recruitment toward Δ*laeA* hyphae. Like the stronger induction of inflammatory cytokines in Fig. 2, the more directed migration of PMNs toward Δ*laeA A. fumigatus*-infected airway lumens compared to WT-infected airway lumens suggests a greater immune response to the Δ*laeA* mutant, which could explain its decreased virulence[24, 28]. Notably, PMNs were not recruited toward spores or fungal germlings in the bronchiole model (Supplementary Fig. 4); while PMNs defend against fungal infection by attacking hyphae, they also have an established role phagocytosing fungal spores[39]. However, it is not yet clear where along the airway this occurs and it may be necessary for PMNs to

be much closer to spores, as would occur in the alveoli, to be able to sense and find this dormant cell type.

**The bronchiole model responds to microbial volatiles**. Besides direct physical contact between pathogen and host, in the lung microenvironment, the exposure to air enables cellular communication via volatile compounds. Bacteria and fungi are adept at producing environment-specific volatiles to disrupt or slow down competing organisms, and in complex multimicrobial communities, secreted volatiles can be used for interkingdom communication[40–42]. Additionally, the host respiratory epithelium can sense and generate an inflammatory response to volatile organic compounds[43, 44]. However, the impact of microbial-derived volatiles on respiratory epithelium remains an open question. The interactions between *A. fumigatus*, *P. aeruginosa*, and host tissue are of particular importance to patients with cystic fibrosis who, when colonized by both microbes, have poorer outcomes[45]. *P. aeruginosa* recovered from patients with cystic fibrosis has been

shown secrete factors that inhibit the growth of *A. fumigatus*[46]. In addition to soluble factor communication, it is reported that *A. fumigatus* and *P. aeruginosa* interact via volatiles to affect microbial growth and even create a unique volatile signature when in coculture[9, 47], though the impact of these volatiles on the human lung remains elusive.

To explore volatile communication between bacteria, fungi, and the organotypic human bronchiole model, we present a microbial culture insert that can be prepared ahead of time and placed into the organotypic bronchiole culture dishes (Fig. 4a, Supplementary Data 2). With a tight lid, the dishes contain volatiles released from the microbial cultures in the insert and allow them to be presented to the air-exposed surface of the bronchiole model (Fig. 4a, b). The insert is engineered to fit snugly within a 50 mm glass bottom dish and can accommodate two independent microbial culture zones, each in the form of a channel wrapping halfway around the insert (Supplementary Fig. 5, Fig. 4b). Additionally, the culture channels are designed to fill by capillary flow, which allows for simple, even filling with a standard micropipette[48]. The response of the bronchiole model to volatile contact with the microbial populations contained in the insert is measured by recovering media from the side lumens for chemical analysis (Fig. 4a).

We measured the inflammatory cytokine response of the bronchiole model as a whole to volatile contact with *A. fumigatus* (WT and Δ*laeA*), *Pseudomonas aeruginosa*, and the coculture of both microbes. The cytokines recovered and measured from the side lumens may be produced directly from the lining endothelial cells or indirectly by fibroblasts and volatile-exposed epithelial cells, which can then diffuse into the side lumens. In general, the inflammatory cytokine response is greater to multimicrobial culture than monomicrobial. GM-CSF and IL-8 are both induced in conditions containing both fungus and bacteria, while levels are minimally changed in conditions with only one microbe (Fig. 4c, d). The responses of IL-6 and IFN-γ are similar with increased production when WT *A. fumigatus* is cocultured with *P. aeruginosa* (Fig. 4e, f). However, the combination of Δ*laeA A. fumigatus* and *P. aeruginosa* has minimal effect on the levels of IL-6 and IFN-γ (Fig. 4e, f). On the other hand, IL-1β is induced in all microbial conditions, though the induction is somewhat stronger when WT *A. fumigatus* is present (Fig. 4g). Other cytokines measured in these experiments are presented in Supplementary Fig. 6.

The bronchiole model responds with a greater inflammatory response when in volatile contact with cocultures of *A. fumigatus* and *P. aeruginosa* compared to either monoculture, suggesting that while the genetic defect in cystic fibrosis predisposes to chronic inflammation, even healthy pulmonary cells generate more inflammation in response to coinfection with *A. fumigatus* and *P. aeruginosa*. Additionally, the changes in proinflammatory cytokines IL-6 and IFN-γ indicate a critical role for *A. fumigatus* secondary metabolism in the bronchiole response to volatile communication with *A. fumigatus* and *P. aeruginosa* cultures: in combination with *P. aeruginosa*, WT *A. fumigatus* strongly induces these cytokines while Δ*laeA A. fumigatus* does not. As LaeA globally regulates *A. fumigatus* secondary metabolite production, including some that are predicted to be volatile[24, 49], this suggests that fungal-derived secondary metabolites mediate the bronchiole model response to indirect contact with *A. fumigatus* and *P. aeruginosa*.

## Discussion

We introduce an organotypic model of the human terminal bronchiole to study host–pathogen interactions at the lung interface. Our model is biologically complex, incorporating three host cell types and up to two microbial populations, yet remains simple to implement and interrogate. We engineered a clickable insert for microbial culture that enables the organotypic bronchiole to communicate with pathogenic microbial populations solely via volatile compounds. With this insert, our platform can investigate and compare both direct physical contact between host and pathogen, as well as volatile contact between host, fungi, and bacteria. The microscale nature of the model allows for recreation of the geometry of the in vivo terminal bronchiole, and enables the use of primary human cells to move closer to the in vivo biology. Further, we find that our model recapitulates the sensitivity of established animal models of invasive aspergillosis (murine and zebrafish) by detecting altered immune responses to the Δ*laeA* fungal mutant, a less virulent strain compared to the WT pathogen[24, 28], including induction of a greater inflammatory cytokine response, and more directed recruitment of PMNs to the site of infection. The differences in immune response occurred in monomicrobial culture (Figs. 2 and 3) and polymicrobial venues (Fig. 4). Importantly, we demonstrate that the volatile interactions between *A. fumigatus* and the organotypic bronchiole model are changed when they are also in volatile contact with *P. aeruginosa*, expanding the use of our platform to advance insights into the multikingdom (bacterial–fungal–human) complexities of human microbiome and infectious diseases.

In addition, the organotypic bronchiole model is compatible with both chemical and biological assays; even operating at microscale, inflammatory cytokines are recovered in sufficient quantity for targeted, accurate measurement using standard kits. The model also supports tracking and measuring leukocyte recruitment, a functional, integrated response to the milieu created by all cell types, human and microbial, contained within the model. Two challenges of working with the microscale bronchiole model include inability to resample model compartments, which makes time series experiments technically difficult, and no extension yet to deliver airborne microbes to the air-filled bronchiole lumen; while we introduce pathogens in suspension or via volatiles, some pulmonary infections occur as a direct result of inhaling airborne organisms. However, our results demonstrate that the organotypic bronchiole model and microbial volatile extension can be used to investigate the initial events in the host response to respiratory pathogens. The versatility of our approach for studying signaling during direct infection as well as signaling via volatiles across kingdoms improves our ability to parse microbiome diseases by systematically investigating how populations of microbes interact with each other and with host cytokine and cellular defenses to cause disease.

## Methods

**Organotypic bronchiole device construction.** Soft lithography was used to fabricate two master molds for the top and bottom halves of the µALI device, as previously described[13]. Briefly, layers of SU-8 100 (Microchem, Newton, MA) were spin-coated onto the wafers, soft-baked at 65 °C for 30–40 min and then 95 °C for 90–120 min, depending on layer thickness. Wafers were then exposed to UV through a mask of the desired pattern followed by post-baking at 95 °C for 20–30 min. After developing in propylene glycol monomethyl ether acetate (Sigma-Aldrich, St. Louis, MO), polydimethylsiloxane (PDMS, Sylgard 184 Silicon Elastomer Kit, Dow Corning Corporation, Midland, MI) was applied to the masters at a ratio of 10:1 base to curing agent and allowed to polymerize for 4 h at 80 °C. Both PDMS layers were then soaked in 70% ethyl alcohol overnight to remove any unreacted PDMS monomers, aligned together, and PDMS rods of the appropriate diameter, 280 µm for side lumens and 550 µm for the center lumen, were inserted, as previously described[15]. Prior to cell culture, devices were plasma-bonded to glass-bottom dishes (MatTek corporation, Ashland, MA), UV-sterilized, and coated to promote collagen adhesion with 2% polyethyleneimine (PEI, Sigma-Aldrich, St. Louis, MO) for 10 min and 0.4% glutaraldehyde (GA, Sigma-Aldrich, St. Louis, MO) for 30 min after which the devices were rinsed with water.

**Collagen gel preparation.** High concentration rat tail collagen I (354249, Corning Life Sciences, Corning, NY) was neutralized on ice with 10× PBS and 1 N sodium

hydroxide (S318, Fisher Scientific, Pittsburgh, PA) and pH was adjusted, if necessary, to 7.2–7.4. Pulmonary fibroblasts suspended in EMEM were added along with fibrinogen (Sigma-aldrich, St. Louis, MO) to achieve a final gel containing 6 mg/mL collagen I, 2 mg/mL fibrinogen, and fibroblasts at 1e6 cells/mL. This solution was added to the gel-loading ports of the µALI device and allowed to polymerize in a 37 °C incubator for 1 h. The PDMS rods were then carefully removed with tweezers leaving patent lumens behind.

**Human cell culture and seeding**. Prior to seeding in the µALIs, normal pulmonary fibroblasts (NPF, CCL-210, ATCC, Manassas, VA) were cultured in Eagle's Minimal Essential Medium (EMEM, 30-2003, ATCC, Manassas, VA) supplemented with 10% fetal bovine serum (FBS, 35-010-CV, Corning-cellgro, Manassas, VA) up to passage 4. Human lung microvascular endothelial cells (LMVEC, CC-2527, Lonza, Basel, Switzerland) were cultured in microvascular endothelial cell growth media (EGM2-MV Bulletkit, CC-3202, Lonza, Basel, Switzerland) up to passage 7. Primary human bronchial epithelial cells (HBEC) were harvested from residual surgical specimens from three healthy lung donors. The Institutional Review Board at the University of Wisconsin waived the need for consent for this protocol as donors were deceased. HBEC were cultured in bronchial epithelial growth medium (BEGM Bulletkit, CC-3170, Lonza, Basel, Switzerland) and were passaged once before use in the µALI. NPF cells were embedded in the collagen by mixing into the collagen immediately prior to filling the devices for a final concentration of 1e6 cells/mL. LMVECs were harvested and resuspended at a concentration of 2.5e7 cells/mL in EGM2-MV before 1 µL was added to each of the side lumens. HBECs were harvested and resuspended at a concentration of 2e7 cells/mL in BEGM before 1.5 µL was added to each of the center lumens. The devices were immediately incubated in a 37 °C incubator for 25 min, then flipped upside down, and to either side each for another 25 min to achieve a uniform coating of the lumen. Non-adherent cells were then rinsed out with fresh media. After two days of culture in growth media, all lumens were given bronchial epithelial differentiation medium (PneumaCult-ALI medium, 05001, STEMCELL Technologies, Vancouver, Canada) for 24 h after which the center, bronchial epithelial lumen was exposed to air. Differentiation media in the side lumens was refreshed daily.

**Staining and imaging**. Dextran permeability assays were performed by adding 4 kDa FITC-labeled dextran (Sigma-Aldrich, St. Louis, MO) to the center of the lumen at 2 mg/mL in media. After a 25 min incubation, the lumens were imaged on a Nikon Ti microscope with NIS Elements software. For immunostaining, lumens were fixed with 4% (vol/vol) paraformaldehyde (30525-89-4, Alfa Aesar, Haverhill, MA), rinsed, and incubated with 0.1 M glycine (G48-500, Fisher Scientific, Pittsburgh, PA) to reduce collagen autofluorescence. Cells were permeabilized with 0.2% Triton-X 100 (vol/vol) (807423, MP Biomedicals, Santa Ana, CA), blocked with 3% (wt/vol) BSA (A9056, Sigma-Aldrich, St. Louis, MO) overnight at 4 °C and then incubated with primary antibody (ab112068 or ab9498, Abcam, Cambridge, UK) diluted 1:25 in 3% BSA for 2 days at 4 °C. Lumens were thoroughly rinsed with 0.1% (vol/vol) Tween (AC278630010, Fisher Scientific, Pittsburgh, PA) over the course of 2 days at 4 °C. Secondary antibodies (A-31571 or A-11001, Invitrogen, Carlsbad, CA) were then added at 1:100 in 3% BSA and incubated for 2 days at 4 °C. The immunostained lumens were again rinsed with 0.1% Tween and finally incubated with DAPI (D3571, Thermo Fisher, Waltham, MA). Images were captured on a Nikon Eclipse Ti (Melville, NY) microscope and background corrected using Image J. Cell viability analysis was performed using calcein AM (C3100MP, Life Technologies), Propidium Iodide (PI) (P3566, Thermo Fisher), and Hoechst (H3570, Thermo Fisher) at 4 and 18 h post inoculation. The ratio of the PI and Hoechst nuclear signals were quantified from microscopy images using JEX (Warrick and Berthier, JEX, available at https://github.com/jaywarrick/JEX. (Accessed: 6 August 2017)). Expectation maximization clustering was performed on the log ratio data to identify two clusters, live and dead. The % viable is calculated as (nuclear—dead).

**Microbial volatile coculture**. Strains of *A. fumigatus* (WT, Af293, from CBS, Centraalbureau voor Schimmelcultures Fungal Biodiversity Centre of the Royal Netherlands Academy of Arts and Sciences; ∆*laeA*, TJW54.2[24]) were maintained as glycerol stocks. They were activated by culture on solid glucose minimal medium (GMM) for 2–3 days at 37 °C and spores were harvested in 0.01% Tween 80. Low-gelling temperature *Aspergillus* minimal media (AMM) was made using 0.7% low-gelling temperature agar (CAS #39346-81-1), 1% glucose, 10 mM ammonium tartrate, and 1X Hutner's trace elements[50]. For experiments with fungal germlings, spores were inoculated into RPMI (11875093 Thermo Fisher) + 10% FBS and placed on a rotary shaker at 250 rpm at 37 °C for 5 h prior to use. Bacteria, *P. aeruginosa* (WT, PA 14, from Dr. Yun Wang, Northwestern University), were grown by culturing for 16–20 h in Luria–Bertani medium at 37 °C in an orbital shaker at 250 rpm. For the volatile microbial cocultures, microbial culture inserts were milled as previously described[51]. Briefly, 4 mm thick polystyrene sheets (ST313400, Goodfellow, Coraopolis, PA) were cut using a CNC micromilling machine (PCNC 700, Tormach, Waunakee, WI). A volume of 60 µL of AMM was pipetted into each side of the microbial culture ring and allowed to solidify at room temperature for 10 min. Then, 15 µL of either fungal spores (1e5 cells/mL) or bacteria (OD = 0.001) in liquid AMM was flowed over the surface of the solid media. Cultures were kept in a

humidified chamber at 37 °C for 4 days at which point the inserts were transferred to the dishes containing the µALI devices, bringing the microbial cultures into volatile contact with the bronchial epithelial cells. After 24 h of volatile coculture, the media was harvested from the side lumens for cytokine analysis.

**Direct microbial coculture**. Direct inoculation of the epithelial center lumens was performed by spinning down the *A. fumigatus* spores and resuspending them in serum-free EGM2-MV to a final concentration of 1e6 cell/mL. Spore suspension (2 µL) was added to each center lumen, and devices were incubated at 37 °C for 18 h prior to analysis at which point hyphae extended from the center lumen out toward the side lumens. Basolateral cytokine production was assayed in the media contained in the side lumens, and neutrophil recruitment to the hyphae was measured.

**Cytokine quantification**. Cytokines present in conditioned media were analyzed using a subset of the HCYTOMAG-60K immunology multiplex assay panel (Millipore, Billerica, MA). Five-microliter samples and standards were incubated with proportionally concentrated beads, and subsequent sample preparation was performed according to manufacturer guidelines. Samples were run in a 96-well plate format through the MagPix system (Luminex Corp., Austin, TX) and data were collected with xPonent software (Luminex). Measured cytokine levels were averaged across two technical replicates and nine biological replicates performed over three separate days.

**PMN migration in organotypic bronchioles**. Migration assays were performed using devices with side lumens that had been seeded with LMVECs at least 2 days prior. Polymorphonuclear cells (PMNs) were obtained from healthy donors in a protocol approved by the University of Wisconsin-Madison Institutional Review Board. Informed written consent was obtained from each subject prior to participation. Purification of PMNs was performed using heparinized blood that was centrifuged over a Percoll gradient (1.090 g/mL), after which red blood cells were lysed with water for 25 s and the PMNs were recovered (PMN fraction was found to be approximately 90–95% neutrophils and 5–10% eosinophils). PMNs were diluted to 4.3e6/mL in Hank's balanced salt solution (HBSS), mixed with Hoechst at 50 µg/mL, and 1 µL of cell suspension was added to each side lumen. Side lumens were imaged in three z-planes at ×4 magnification on an Olympus IX-81 optical microscope with Slidebook capture software (3i, Denver, CO) at intervals of 3–5 min for 3 h. In MATLAB (MathWorks, Natick, Ma), the z-planes were flattened into a single image, brightness was corrected, and the images were denoized. Cells were identified in each frame using MATLAB's imfindcircles algorithm and tracked using u-track 2.0[52]. Migrating cells were considered to be those present for 4 or more continuous frames and that traveled out of the lumen for a total distance of at least 150 µm. Direction of migration, toward or away from the center lumen, was noted.

**Statistical analysis**. GraphPad Prism 6 software (La Jolla, CA) was used for statistical analysis. All tests were performed using either unpaired, two-tailed Student's *t*-test (Supplementary Fig. 2c, e), ordinary two-way ANOVA with Tukey's multiple comparisons test (Fig. 2d), or ordinary one-way ANOVA with Tukey's multiple comparisons test (Figs. 2e–g, 3d, and 4c–g, Supplementary Figs. 3a–e, 4, and 6a–c). When necessary, data were log-transformed to better satisfy the assumptions of ANOVA.

**Data availability**. All relevant data supporting the findings of the study are available in this article and its Supplementary Information files, or from the corresponding authors upon request.

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

## Acknowledgements

We thank Karina Lugo Cintron for assistance with immunofluorescent staining, Mary Regier and Heidi Schoen for help with the brainstorming sessions and thoughtful discussions, and Samantha Wingett for help with histological processing. We also thank Paul Fichtinger for help with PMN purification and Benjamin Horman for assistance with device fabrication. This work was supported in part by the National Science Foundation-Emerging Frontiers in Research and Innovation-MIKS: Grant 1136903 (D.J.B., N.P.K., and E.B.); in part by the National Institute of Health: 4R01 CA155192 (Y.R.A.G.), R01 EB010039 (J.A.J.T.), P01 HL070831 (R.A.B.S., J.E.G.), U19 AI104317 (J.E.G.), R01 HL115118 (L.C.D.), K12 DK100022 (A.B.T.), R01 AI065728 (N.P.K.), and P30 CA014520 (J.A.J.T. and D.J.B.); in part by the National Library of Medicine: 5T15LM007359 (L.J.B.); in part by the University of Wisconsin Graduate Engineering Research Scholars (J.A.J.T.); and in part by the University of Washington (A.B.T.). L.J.B. is a student in the UW Medical Scientist Training Program (T32 GM008692). J.A.J.T. is a UW Hematology trainee (T32 HL07899).

## Author contributions

L.J.B., M.N., J.A.J.T., L.C.D., R.A.B.-S., J.E.G., A.B.T., N.P.K., E.B., and D.J.B. designed the study. L.J.B., C.L.P., Y.R.A.-G., M.N., and J.A.J.-T. performed experiments. L.J.B., A.B.T., N.P.K., E.B., and D.J.B. analyzed data. L.J.B., C.L.P., Y.R.A.-G., A.B.T., N.P.K., E.B., and D.J.B. wrote the manuscript.

## Additional information

**Competing interests:** J.A.J.T. holds equity in Onexio Biosystems LLC. A.B.T. holds equity in Stacks to the Future LLC. E.B. is VP of R&D and Clinical Affairs at Tasso Inc. and holds equity in Stacks to the Future LLC. D.J.B. holds equity in Bellbrook Labs LLC, Tasso Inc., Stacks to the Future LLC, Lynx Biosciences LLC, Onexio Biosystems LLC and Salus Discovery LLC. The remaining authors declare no competing financial interests.

