## [Peer Review File · Nature Communications]

Reviewers' comments:

Reviewer #1 (Remarks to the Author):

Age and sex of primary cell donors should be listed.

Viability of cultures (live/dead) cell assay should have been done. Cell loss is part of the response to pathogens such as *Aspergillus fumigatus* and *Pseudomonas aeruginosa* and would be critical information if you are developing this as a standardized pathogen model. I hope this is your goal as it has all of the elements needed for use as an upper airway model system.

AMM were grown into a microbial culture ring. Spores or organisms were not airborne or aerosol delivered changing how cells come into contact with pathogen. Route of pathogen administration is always important and providing pathogens in their normal route for transmission would strengthen your experimental design. Despite this issue this was one of the best respiratory pathogen models I have seen so far!

Neutrophil migration and response was strength in the experimental design. Non autologous macrophages can be added to cultures using the same protocol to allow for innate immune response to develop.

Supplemental data regarding cytokine production should have been a figure in the manuscript and not supplemental. While I agree that design of the device is important the response of the device should also be highlighted. That is what validates your model as a pathogen/microbiome model rather than just an airway model.

I recommend accepting this manuscript for publication once sex/age of the donors has been added and a statement regarding how the route of microbe exposure simulates or does not simulate exposure routes for these microbes in natural infection.

Reviewer #2 (Remarks to the Author):

The organotypic model presented by Barkal et al is original but the data do not really support the excitement that this reviewer was expecting from such a technological development. Such a complicate miniaturized device is not needed to follow the neutrophil migration or quantify the differences in cytokine responses between a mutant and its parental strain. In addition, most of the data shown here have been already published and the ms just confirmed results already obtained. It would have been more interesting to see new data explaining the virulence of *A.fumigatus* which could have been only demonstrated by the use of this new device. In addition, the pathogenicity of *A.fumigatus* is not due so much to the fungus itself but to the severe immunosuppression of the host. Can the device be used here to mimick immunosuppression and follow the fungal growth under these conditions and what can we learn from such an experiment (which would be different from the current knowledge)? This would be a true advance in the study of *A.fumigatus*. And this would be more important than follow defects in PMN migration since there are no (or a very low number of) neutrophils in immunosuppressed patients at risk for Aspergillosis. More interesting than following the effect of the volatome on the production of classical cytokines, it would be more important to study the effect of the contact between bacteria and fungus on the epithelium and follow the damages induced by these opportunistic pathogens during joint infections. Is it possible to follow in the device two pathogens with different growth rate and different ways of locomotion. If this is possible, then again it would be a real advance.

Reviewer #3 (Remarks to the Author):

The study by Barkal et al investigates the suitability of a new microscale organotypic human bronchiole model for investigating virulence of the pathogenic fungus *Aspergillus fumigatus*. Furthermore, evidence is shown for the use of the system to investigate the effect of volatiles produced by microbial communities on immune responses.

The new models system adds several levels of complexity to traditional in vitro immune interaction studies by resembling a three-dimensional lung environment with different cell types and the ability to add various immune effector cells. Despite its rather complex preparation, this model system appears to bridge a gap between traditional in vitro immune cell-pathogen interactions and murine infection models. Given the fact that generating this system requires high technological knowledge combined with the availability of human primary donor cells, this system appears not suitable for high throughput screenings of fungal mutants. However, it may be highly suitable to monitor immune interactions of pre-selected mutants that displayed phenotypes in alternative infection models.

The authors established their system by investigating the immune reactions caused by an *Aspergillus fumigatus* wild-type strain against a *laeA* mutant. In addition, the effect of volatiles on epithelial cells was studied by investigating combinations of *A. fumigatus* and the bacterial lung pathogen *Pseudomonas aeruginosa*. While all studies appear technically sound, there are some questions that should be addressed.

In the cytokine release studies after fungal infection, a time point of 18 h has been selected at which fungal mycelium has already been well developed in the artificial airways. Why has this time point been selected? Is it possible to remove samples at different time points and repeatedly from the model? The ability to investigate disease progression and its accompanied cytokine response would add much to the study. Otherwise, only snapshots can be taken, which is also a drawback in conventional murine infection models. In this respect, either data for different time points should be added or, at least, this drawback addressed in the discussion.

What is the rationale to select the *delta_laeA* strain in this comparative analysis? Although the authors have a long-standing experience with this mutant, its phenotype is extremely complex ranging from alteration of the secondary metabolite profile (good for volatile studies) to developmental and other phenotypic defects. Therefore, it is difficult to correlate changes in the cytokine pattern to a specific defect. Furthermore, although the *laeA* mutant had been tested in a murine infection model for its virulence and displayed a strong virulence attenuation, no data on cytokine levels have been collected in the murine infection model. Therefore, it appears difficult to judge the data on cytokine levels that have been obtained here. However, if cytokine levels from a murine infection model are available, these should be presented.

Similarly, the rationale for adding neutrophils in the leucocyte migration assay when hyphae of fungi have already well developed (at 18 h) is difficult to understand. In a non-neutropenic infection model neutrophils are generally recruited to the site of infection within a few couple of hours supporting the inactivation of germinating conidia. The *laeA* mutant has been shown to be more rapidly phagocytosed by macrophages and is much less effective in inactivating PMNs (Bok et al., 2005, *Eukaryotic Cell*, 4:1574-1582). Therefore, showing in this model system that early addition of PMNs is able to restrict growth of *laeA* mutant but possibly not of wild-type conidia would have strongly increased the value of the study.

The volatile model shows very interesting results as it clearly shows that microorganisms interact in terms of volatile production and that this influences the immune response of the infected host. However, is it possible to recover the gas phase from this model for subsequent volatile analyses? If so, this should be stated here. Furthermore, are there any ideas on the volatiles that may be

produced and are responsible for the observed effects?

Minor comments:

Line 386-387: Please modify this sentence. What do you mean by ... interact with varied host defences to cause disease?

Line 426 and line 497. It is a bit confusing to read that informed written consent from each subject prior to participation (line 497) was obtained, when line 426 states that the Review board waived the need for consent for this protocol. What was the reason for waiving this and why has it done anyhow?

Line 448: I assume that Triton X-100 has been used.

Line 454: himmunistained may read immunostained.

Line 463: I assume that Hutner's trace element solution rather than Hunter's trace elements have been used.

Line 474-475: I cannot see experiments in which PMNs were added in volatile studies. Either show these data or remove the sentence.

Line 491: Cytokine levels were averaged across two technical replicates. What about the biological replicates?

We thank the reviewers for their suggestions and feedback. We have incorporated changes (including additional data) into the manuscript and Supporting Information to address their points. Below we respond to each point (responses are in blue text) and describe how we have modified our manuscript to address the suggestions. We have provided a manuscript with “tracked changes” as well as a clean copy.

Reviewers' Comments to Author

Reviewer #1:

Age and sex of primary cell donors should be listed.

Unfortunately, donor age and sex are not recorded at the time of tissue harvest due to concern for patient information privacy. We do know that the donors had no chronic lung disease, were not infected at the time of death, and were relatively healthy. Using this protocol, we have had donors of both sexes, and donor ages ranging from 5 – 70 years old, with the majority falling from teens – 50s, but we are unsure of which donors specifically contributed tissue to the current study.

Viability of cultures (live/dead) cell assay should have been done. Cell loss is part of the response to pathogens such as *Aspergillus fumigatus* and *Pseudomonas aeruginosa* and would be critical information if you are developing this as a standardized pathogen model. I hope this is your goal as it has all of the elements needed for use as an upper airway model system.

We thank the reviewer for this suggestion. We have performed additional experiments to test viability of the bronchiolar epithelial cells during direct infection with *A. fumigatus* spores, which is now shown in Fig 2d. Fungal infection did not impact viability significantly at 4 hours post infection, when spores are still germinating, or 18 hours post infection, when there is a dense mat of hyphae. However, looking specifically at lumens inoculated with $\Delta laeA$ spores, there is a significant decrease in lumen viability between 4 and 18 hours post infection, suggesting cell loss may play more of a role in later infection by the mutant spores.

AMM were grown into a microbial culture ring. Spores or organisms were not airborne or aerosol delivered changing how cells come into contact with pathogen. Route of pathogen administration is always important and providing pathogens in their normal route for transmission would strengthen your experimental design. Despite this issue this was one of the best respiratory pathogen models I have seen so far!

Thank you for bringing up this important point. We would like to be able to deliver pathogens through air in the future. For now, we have revised the manuscript to make it more clear that pathogens were either directly delivered in liquid or grown at a distance communicating solely via volatiles.

Neutrophil migration and response was strength in the experimental design. Non autologous macrophages can be added to cultures using the same protocol to allow for innate immune response to develop.

This is definitely an area of interest for future investigations.

Supplemental data regarding cytokine production should have been a figure in the manuscript and not supplemental. While I agree that design of the device is important the response of the device should also be highlighted. That is what validates your model as a pathogen/microbiome model rather than just an airway model.

We struggled with organizing figures containing all cytokine data that maintained readability and did not push us over length suggestions. If the editor is flexible on these points, we could certainly bring the rest of the cytokine data back into the main figures.

I recommend accepting this manuscript for publication once sex/age of the donors has been added and a

statement regarding how the route of microbe exposure simulates or does not simulate exposure routes for these microbes in natural infection.

Reviewer #2:

The organotypic model presented by Barkal et al is original but the data do not really support the excitement that this reviewer was expecting from such a technological development. Such a complicated miniaturized device is not needed to follow the neutrophil migration or quantify the differences in cytokine responses between a mutant and its parental strain. In addition, most of the data shown here have been already published and the ms just confirmed results already obtained. It would have been more interesting to see new data explaining the virulence of *A.fumigatus* which could have been only demonstrated by the use of this new device.

Thank you for bringing up this important point. While neutrophil migration to fungi can be studied in simpler devices, the model presented here is focused on providing a more relevant microenvironment. To our knowledge, there are no published neutrophil migration experiments that take place in the presence of the epithelial cells you would expect to be present in vivo. The bronchiole model allows us to investigate the role of epithelial cells in sounding the alarm and secreting the factors necessary to prime the endothelial cell lumens to allow for PMN extravasation. The integrated model also allows us to measure complimentary responses including both migration and cytokine production. In developing this new, complex model, we worked to strike a balance between validation (recapitulation of cytokine responses expected from other published literature) and novel studies (the volatile work is without precedent).

In addition, the pathogenicity of *A.fumigatus* is not due so much to the fungus itself but to the severe immunosuppression of the host. Can the device be used here to mimick immunosuppression and follow the fungal growth under these conditions and what can we learn from such an experiment (which would be different from the current knowledge)? This would be a true advance in the study of *A.fumigatus*. And this would be more important than follow defects in PMN migration since there are no (or a very low number of) neutrophils in immunosuppressed patients at risk for Aspergillosis. More interesting than following the effect of the volatome on the production of classical cytokines, it would be more important to study the effect of the contact between bacteria and fungus on the epithelium and follow the damages induced by these opportunistic pathogens during joint infections. Is it possible to follow in the device two pathogens with different growth rate and different ways of locomotion. If this is possible, then again it would be a real advance.

We drew from the knowledge that neutropenic hosts are particularly susceptible to *A. fumigatus* infection with the idea that if neutropenia is a significant risk factor, neutrophils must play a critical role in responding to *A. fumigatus* invasion. By characterizing the neutrophil response to invasive *A. fumigatus* infection, we establish a normal baseline from which we can study how the dysfunctional neutrophils of neutropenic hosts or patients on corticosteroid therapy have a different response, though that is outside the scope of this manuscript. Additionally, our neutrophil migration experiments allowed us to investigate fungal factors that might contribute to pathogenicity by isolating changes to a fungal mutant. With our model, investigating the effects of neutropenia and fungal factors are both possible in future studies. We also thank you for the suggestion of looking at direct contact co-infection with *A. fumigatus* and *P. aeruginosa*, it is definitely of interest for our future work.

Reviewer #3:

The study by Barkal et al investigates the suitability of a new microscale organotypic human bronchiole

model for investigating virulence of the pathogenic fungus *Aspergillus fumigatus*. Furthermore, evidence is shown for the use of the system to investigate the effect of volatiles produced by microbial communities on immune responses.

The new models system adds several levels of complexity to traditional in vitro immune interaction studies by resembling a three-dimensional lung environment with different cell types and the ability to add various immune effector cells. Despite its rather complex preparation, this model system appears to bridge a gap between traditional in vitro immune cell-pathogen interactions and murine infection models. Given the fact that generating this system requires high technological knowledge combined with the availability of human primary donor cells, this system appears not suitable for high throughput screenings of fungal mutants. However, it may be highly suitable to monitor immune interactions of pre-selected mutants that displayed phenotypes in alternative infection models.

The authors established their system by investigating the immune reactions caused by an *Aspergillus fumigatus* wild-type strain against a *laeA* mutant. In addition, the effect of volatiles on epithelial cells was studied by investigating combinations of *A. fumigatus* and the bacterial lung pathogen *Pseudomonas aeruginosa*. While all studies appear technically sound, there are some questions that should be addressed.

In the cytokine release studies after fungal infection, a time point of 18 h has been selected at which fungal mycelium has already been well developed in the artificial airways. Why has this time point been selected? Is it possible to remove samples at different time points and repeatedly from the model? The ability to investigate disease progression and its accompanied cytokine response would add much to the study. Otherwise, only snapshots can be taken, which is also a drawback in conventional murine infection models. In this respect, either data for different time points should be added or, at least, this drawback addressed in the discussion.

Thank you for this suggestion. We selected the 18 hour time point because this represented an established infection; at 18 hours, the spores have germinated and hyphae extend nearly to the side lumens. While there are most certainly interesting observations to be made at earlier time points, we were limited by the cytokine assay, which requires full harvest of the media from the side lumens. The model cannot be resampled; given the low volume of the model, complete removal of media from the side lumens is expected to significantly influence the biology of the system. We have added discussion of this drawback to the manuscript in the conclusion.

What is the rationale to select the Δ *laeA* strain in this comparative analysis? Although the authors have a long-standing experience with this mutant, its phenotype is extremely complex ranging from alteration of the secondary metabolite profile (good for volatile studies) to developmental and other phenotypic defects. Therefore, it is difficult to correlate changes in the cytokine pattern to a specific defect. Furthermore, although the *laeA* mutant had been tested in a murine infection model for its virulence and displayed a strong virulence attenuation, no data on cytokine levels have been collected in the murine infection model. Therefore, it appears difficult to judge the data on cytokine levels that have been obtained here. However, if cytokine levels from a murine infection model are available, these should be presented.

We chose the *laeA* mutant primarily for its characterized decrease in virulence and unpublished data we had suggesting it was altered in volatile production. We were also excited to use the *laeA* mutant in the volatile studies given its wide-reaching secondary metabolite changes. In sum, we think the choice of the *laeA* mutant was a good choice for this first manuscript. Other mutants certainly would be possible to use and now that we know that the model is responsive to gross changes in fungal phenotype, we can proceed with other mutants to investigate specific fungal defects. We do not have cytokine levels from murine infection with the *laeA* mutant.

Similarly, the rational for adding neutrophils in the leucocyte migration assay when hyphae of fungi have

already well developed (at 18 h) is difficult to understand. In a non-neutropenic infection model neutrophils are generally recruited to the site of infection within a few couple of hours supporting the inactivation of germinating conidia. The *laeA* mutant has been shown to be more rapidly phagocytosed by macrophages and is much less effective in inactivating PMNs (Bok et al., 2005, *Eukaryotic Cell*, 4:1574-1582). Therefore, showing in this model system that early addition of PMNs is able to restrict growth of *laeA* mutant but possibly not of wild-type conidia would have strongly increased the value of the study. In the work presented in our manuscript, we focus on neutrophil response to hyphae as this is one of their distinct functions in the response to fungal infection (1). Clinically, it is observed that neutropenic hosts suffer extensive tissue invasion whereas patients on prolonged courses of corticosteroids (who have adequate numbers of neutrophils but with aberrant function) develop aspergillomas, fungal balls with minimal tissue invasion. These clinical observations support that neutrophils play a role in defending against hyphae and tissue invasion, not just early killing of spores.

Still, to investigate early PMN response to infection in the organotypic bronchiole model, we performed an additional migration experiment shown in Supplementary Fig. 4. In this new experiment, we investigate the potential of PMNs to migrate toward center lumens filled with either fungal spores that have not yet germinated, or fungal germlings midway through the germination process. We observe that while there is a trend toward more directional migration of PMNs in models infected with fungal spores or germlings, especially the *dlaeA* mutant, these conditions are not significantly different.

In future work, we would like to incorporate macrophages, which both play their own role in early host defense against fungal spores and may also help drive, through signaling crosstalk, early recruitment of neutrophils (2).

1. Gazendam, R. P., van Hamme, J. L., Tool, A. T. J., Hoogenboezem, M., van den Berg, J. M., Prins, J. M., et al. (2016). Human Neutrophils Use Different Mechanisms To Kill *Aspergillus fumigatus* Conidia and Hyphae: Evidence from Phagocyte Defects. *The Journal of Immunology*, 196(3), 1272–1283.
2. Hohl, T. M. (2017). Immune responses to invasive aspergillosis: new understanding and therapeutic opportunities. *Current Opinion in Infectious Diseases*, 30(4), 364–371.

The volatile model shows very interesting results as it clearly shows that microorganisms interact in terms of volatile production and that this influences the immune response of the infected host. However, is it possible to recover the gas phase from this model for subsequent volatile analyses? If so, this should be stated here. Furthermore, are there any ideas on the volatiles that may be produced and are responsible for the observed effects?

Recovering the gas phase of an individual device would be technically challenging and limit our ability to assign volatiles a specific compartment of origin, but more generally, volatile organic compounds of microbial origin are traditionally identified by scraping fungal and bacterial growth from the devices into glass sampling containers, followed by solid phase microextraction mass-spectrometry gas chromatography (1, 2). The same process should be feasible using the culture inserts, and microbes from multiple inserts may be pooled to obtain metabolite concentrations above the limit of detection; this is an area of interest for our future studies.

Perhaps the bioactive volatiles are some of those reported by Briard et al. in their *A. fumigatus* and *P. aeruginosa* volatile coculture experiments (3).

1. Heddergott C, Calvo AM, Latgé JP. The Volatome of *Aspergillus fumigatus*. *Eukaryotic Cell*. 8, 1014-25 (2014).
2. Polizzi V, Adams A, De Saeger S, Van Peteghem C, Moretti A, De Kimpe N. Influence of various growth parameters on fungal growth and volatile metabolite production by indoor molds. *Science of the Total Environment*. 414, 277-86 (2012).

3. Briard, B., Heddergott, C. & Latgé, J.-P. Volatile Compounds Emitted by *Pseudomonas aeruginosa* Stimulate Growth of the Fungal Pathogen *Aspergillus fumigatus*. *mBio* 7, e00219 (2016).

Minor comments:

Line 386-387: Please modify this sentence. What do you mean by ... interact with varied host defences to cause disease?

Thank you for this suggestion. By varied host defenses, we meant cytokine responses and immune cells responses as investigated in the manuscript. This has been modified in the text.

Line 426 and line 497. It is a bit confusing to read that informed written consent from each subject prior to participation (line 497) was obtained, when line 426 states that the Review board waived the need for consent for this protocol. What was the reason for waiving this and why has it done anyhow?

Informed consent was gathered for use of primary PMNs. It was not needed and therefore not gathered from epithelial cell donors, as these donors were deceased. This has been clarified in the text.

Line 448: I assume that Triton X-100 has been used.

Thank you for catching this – it has been clarified

Line 454: himmunistained may read immunostained.

Thank you for catching this – it has been corrected

Line 463: I assume that Hutner's trace element solution rather than Hunter's trace elements have been used.

Thank you for catching this – it has been corrected

Line 474-475: I cannot see experiments in which PMNs were added in volatile studies. Either show these data or remove the sentence.

Thank you for catching this – it has been corrected

Line 491: Cytokine levels were averaged across two technical replicates. What about the biological replicates?

This has been clarified

REVIEWERS' COMMENTS:

Reviewer #1 (Remarks to the Author):

Overall this is a well written ,manuscript. The focus seems to be on the device development and validation of the device rather than the use of the model to examine host-pathogen interactions. One issue I have is the use of the volatile-mediated communication between the pathogen and the host (model system). Generally we refer to airborne materials as airborne materials. This refers to droplets, aerosols and particulate matter. Volatile refers to readily vaporizable at a relatively low temperature. Volatile materials and airborne materials do not really mean the same thing and this concept needs to be clarified in the manuscript prior to publication. Volatile organic compounds are different from materials produced by pathogens although release of microbial products and wastes does trigger host responses.

Reviewer #3 (Remarks to the Author):

The manuscript by Barkal et al has greatly improved by the revisions made by the authors. The organotypic lung model is an interesting alternative to murine infection models as different layers of complexity can be introduced. Therefore, its use may not only be limited to the investigation of fungal pathogens, but also other pathogens causing lung disease.

Overall, my concerns have been sufficiently addressed by the authors. However, as a comment on the authors' response concerning the function of neutrophils in controlling fungal infections. I agree that neutrophils attack fungal hyphae and are of tremendous importance to resolve an established infection. However, a recent study published in Science (Shlezinger et al., 2017, Science 357: 1037-1041) clearly showed that the interaction of *A. fumigatus* conidia with neutrophils leads to the induction of an apoptotic process in conidia. As several studies showed the phagocytosis of conidia by neutrophils in murine infection models, controlling the germination of conidia belongs to the main tasks of these immune cells. If space allows, this might be added.

In addition, please revise the following sentences/words:

Line 158: In addition to WT *A. fumigatus*, we measure model response the knockout *A. fumigatus* strain, $\Delta laeA$.  This sentence is difficult to understand although I get the meaning.

Line 170-171: ..., though later infection with fungal mutant $\Delta laeA$ does result in lower viability than early infection (Fig. 2d).  I'm a bit confused here. From my understanding the "later infection" should show data from 4 h whereas the "early infection" should resemble the 18 h post infection. Figure 2d shows the opposite from the description in the text.

Line 442: It is still "Hunter's" trace elements rather than Hutner's.  Not corrected as stated in the response letter.

We thank the reviewers for their suggestions and feedback. We have incorporated changes into the manuscript and Supporting Information to address their points. Below we respond to each point (responses are in blue text) and describe how we have modified our manuscript to address the suggestions. We have provided a manuscript with “tracked changes” as well as a clean copy.

Reviewers' Comments to Author

Reviewer #1:

Overall this is a well written manuscript. The focus seems to be on the device development and validation of the device rather than the use of the model to examine host-pathogen interactions. One issue I have is the use of the volatile-mediated communication between the pathogen and the host (model system). Generally, we refer to airborne materials as airborne materials. This refers to droplets, aerosols and particulate matter. Volatile refers to readily vaporizable at a relatively low temperature. Volatile materials and airborne materials do not really mean the same thing and this concept needs to be clarified in the manuscript prior to publication. Volatile organic compounds are different from materials produced by pathogens although release of microbial products and wastes does trigger host responses.

Thank you for bringing up this important point. We appreciate the distinction between volatiles and airborne particulate matter. In the organotypic bronchiole model, it is highly unlikely that airborne particulates play a role in the coculture experiments. The experimental setup has been designed to avoid physical contamination of the airway lumen with the microbial cultures located 1 centimeter or more from the ports of the air-filled lumens. Additionally, there is high resistance to airflow through the lumens; a very rough back-of-the-envelope calculation comparing the resistance to airflow (which has a radius⁴ term) through a 550 um diameter lumen with a 9.65 mm diameter lumen (a very conservative abstraction of the dish, which has a depth of 9.65 mm) indicates resistance to flow through the lumen would be ~95,000 times greater than that of the rest of the dish. At most, you may anticipate deposition of particulate matter in the ports of the airway lumens. Notably, we did not observe spores/and or fungal growth in the ports as would be expected if spores were airborne and had settled in the ports during the 24 hour experiment. Additionally, it is well known that many microbial secondary metabolites are produced in coculture situations and many secreted microbial metabolites are volatile compounds (1-3). Therefore, we are comfortable using the term volatile to indicate fungal or bacterial secondary metabolites that readily vaporize and mediate the communication we observe between pathogen and our host model.

- 1. Audrain B, Farag MA, Choong-Min R, and Ghigo, JM. Role of bacterial volatile compounds in bacterial biology. FEMS Microbiol Rev. 2015 Mar;39(2):222-33.
- 2. Weigl F, Ghirardo A, Schnitzler JP, and Pritsch K. Sesquiterpene emissions from *Alternaria alternata* and *Fusarium oxysporum*: Effects of age, nutrient availability, and co-cultivation. Sci Rep. 2016 Feb 26;6:22152.
- 3. Bennet JW and Inamdar AA. Are some fungal volatile organic compounds (VOCs) mycotoxins? Toxins (Basel). 2015 Sep 22;7(9):3785-804.

Reviewer #3:

The manuscript by Barkal et al has greatly improved by the revisions made by the authors. The organotypic lung model is an interesting alternative to murine infection models as different layers of complexity can be introduced. Therefore, its use may not only be limited to the investigation of fungal pathogens, but also other pathogens causing lung disease.

Overall, my concerns have been sufficiently addressed by the authors. However, as a comment on the authors' response concerning the function of neutrophils in controlling fungal infections. I agree that neutrophils attack fungal hyphae and are of tremendous importance to resolve an established infection. However, a recent study published in Science (Shlezinger et al., 2017, Science 357: 1037-1041) clearly showed that the interaction of *A. fumigatus* conidia with neutrophils leads to the induction of an apoptotic process in conidia. As several studies showed the phagocytosis of conidia by neutrophils in murine

infection models, controlling the germination of conidia belongs to the main tasks of these immune cells. If space allows, this might be added.

Thank you for making this point and bringing this new paper to our attention. We certainly agree that PMNs are capable of phagocytosing spores; however, it is unclear if this function occurs at the level of alveoli and/or higher up in the airway at the level of the bronchiole (even *in vivo* studies have been done using BAL fluid, which cannot distinguish the two). We have added a brief discussion of the role for PMN phagocytosis to the end of the section on PMN migration.

In addition, please revise the following sentences/words:

Line 158: In addition to WT *A. fumigatus*, we measure model response the knockout *A. fumigatus* strain, $\Delta laeA$.  This sentence is difficult to understand although I get the meaning.

Thank you for pointing this out, it has been rephrased in the text

Line 170-171: ..., though later infection with fungal mutant $\Delta laeA$ does result in lower viability than early infection (Fig. 2d).  I'm a bit confused here. From my understanding the "later infection" should show data from 4 h whereas the "early infection" should resemble the 18 h post infection. Figure 2d shows the opposite from the description in the text.

This sentence has been removed as reanalysis with 2-way ANOVA comparing all conditions to all other conditions simultaneously, with Tukey's multiple comparisons correction, resulted in no statistically significant difference between groups; the prior analysis used two separate ANOVAs, which was incorrect.

Line 442: It is still "Hunter's" trace elements rather than Hutner's.  Not corrected as stated in the response letter.

Apologies, this has been corrected now